# Electric-field control of field-free spin-orbit torque switching via laterally modulated Rashba effect in Pt/Co/AlOₓ structures

Min-Gu Kang[1,8], Jong-Guk Choi[1,8], Jimin Jeong[1], Jae Yeol Park [1], Hyeon-Jong Park[2], Taehwan Kim[3], Taekhyeon Lee[4], Kab-Jin Kim [4], Kyoung-Whan Kim [5], Jung Hyun Oh[6], Duc Duong Viet[7], Jong-Ryul Jeong [7], Jong Min Yuk[1], Jongsun Park[3], Kyung-Jin Lee [4] & Byong-Guk Park [1✉]

Spin-orbit coupling effect in structures with broken inversion symmetry, known as the Rashba effect, facilitates spin-orbit torques (SOTs) in heavy metal/ferromagnet/oxide structures, along with the spin Hall effect. Electric-field control of the Rashba effect is established for semiconductor interfaces, but it is challenging in structures involving metals owing to the screening effect. Here, we report that the Rashba effect in Pt/Co/AlOₓ structures is laterally modulated by electric voltages, generating out-of-plane SOTs. This enables field-free switching of the perpendicular magnetization and electrical control of the switching polarity. Changing the gate oxide reverses the sign of out-of-plane SOT while maintaining the same sign of voltage-controlled magnetic anisotropy, which confirms the Rashba effect at the Co/ oxide interface is a key ingredient of the electric-field modulation. The electrical control of SOT switching polarity in a reversible and non-volatile manner can be utilized for programmable logic operations in spintronic logic-in-memory devices.

[1] Department of Materials Science and Engineering, KAIST, Daejeon 34141, Korea. [2] KU-KIST Graduate School of Converging Science and Technology, Korea University, Seoul 02841, Korea. [3] School of Electrical Engineering, Korea University, Seoul 02841, Korea. [4] Department of Physics, KAIST, Daejeon 34141, Korea. [5] Center for Spintronics, Korea Institute of Science and Technology, Seoul 02792, South Korea. [6] Department of Materials Science and Engineering, Korea University, Seoul 02841, Korea. [7] Department of Materials Science and Engineering, Chungnam National University, Daejeon 34134, Korea. [8]These authors contributed equally: Min-Gu Kang, Jong-Guk Choi. ✉email: bgpark@kaist.ac.kr

Spin-orbit torque (SOT) is a spin torque arising from spin–orbit coupling in heavy metal (HM)/ferromagnet (FM)/oxide structures, in which a spin current generated by the spin Hall effect[1–3] in the HM and the Rashba effect[4–6] at the HM/FM and/or FM/oxide interfaces exerts a torque on the FM and manipulates the magnetization direction. Providing fast and energy-efficient magnetization switching[7–9] and domain wall motion[10,11], SOT is being intensively investigated[12,13] as an alternative technique to manipulate the magnetization for various spintronic devices including magnetic random access memories[8,9,14], spintronic logics[15,16], and oscillators[17,18]. While the spin Hall effect is of bulk origin from the spin–orbit coupling in HM[19], the Rashba effect is of interface origin from the broken inversion symmetry at interfaces in HM/FM/oxide heterostructures[20]. Electrical control of the Rashba effect was demonstrated at semiconductor interfaces;[21-24] however, it has not been clearly reported in metallic structures so far for the following reasons. First, the application of an electric field in the metallic structures is limited to one or two atomic layers due to the Coulomb screening effect. Second, it is difficult to distinguish the changes in the Rashba effect from the concomitant modification of magnetic properties owing to the applied gate voltage. Note that the voltage control of the magnetism has been actively studied[25–29] since it offers the reduction of power consumption for the magnetization switching without degrading the thermal stability. Recent studies demonstrated that the SOT in HM/FM/oxide structures is controlled using a gate voltage[30–32] through modulation of oxygen concentration at the interface by inducing oxygen ion migration. Note that the voltage effect based on oxygen ion migration is large but slow compared to that caused by the charge distribution mechanism[33–35].

In this article, we report that the lateral modulation of Rashba effect in Pt/Co/AlO$_x$ structures by using two side gates generates out-of-plane SOT, allowing electrical control of field-free switching of perpendicular magnetization. In the device configuration illustrated in Fig. 1a, applying different gate voltages to the two side gates induces an additional lateral symmetry-breaking along the $y$-direction. In the presence of a charge current along the $x$-direction, the symmetry analysis[36] shows that the lateral symmetry-breaking along the $y$-direction results in additional SOTs in directions of $\mathbf{m} \times \mathbf{z}$ and $\mathbf{m} \times (\mathbf{m} \times \mathbf{z})$, where $\mathbf{m}$ and $\mathbf{z}$ are unit vectors along the magnetization and the thickness direction, respectively. This out-of-plane SOT or $z$-SOT can switch perpendicular magnetization without an external magnetic field[36–38] and a theoretical study suggests that the $z$-SOT can greatly reduce the switching current[39].

In this work, we show that the $z$-SOT and associated switching polarity are controllable by gate voltage in a reversible and non-volatile manner, offering programmable logic operations in spintronic login-in-memory devices as demonstrated below. Moreover, we provide a microscopic origin of the $z$-SOT induced by asymmetric gate voltages. It is noted that the above symmetry argument is valid regardless of the source of lateral symmetry-breaking along the $y$-direction. However, it is of crucial importance to identify the microscopic origin of the electric-field-induced lateral symmetry breaking for further improvement of device performance. To this end, we show that the $z$-SOT depends on gate oxide materials; the direction of the $z$-SOT in the identical Pt/Co/AlO$_x$ structures is reversed by changing the gate oxide from TiO$_2$ to ZrO$_2$. The two gate oxides exhibit opposite electric-field effects on the potential barrier; for TiO$_2$ (ZrO$_2$), a positive voltage decreases (increases) the potential barrier height, resulting in the modification of the built-in electric field and associated Rashba effect at the Co/oxide interface. This, together with the same sign of voltage-controlled magnetic anisotropy effect regardless of the gate oxide, suggests that the $z$-SOT is

mainly due to the lateral modulation of the Rashba effect at the Co/AlO$_x$ interface, which is further supported by the measurements of the gate-voltage dependence of field-like SOTs.

## Results

**Electric-field control of deterministic spin–orbit torque switching.** To demonstrate the $z$-SOT generated by asymmetric gate voltages, we fabricate Pt (5 nm)/Co (1.4 nm)/AlO$_x$ (2 nm) Hall-bar devices, in which two side gates consisting of a gate oxide of TiO$_2$ (40 nm) and a gate electrode of Ru (50 nm) are integrated (Fig. 1b and Methods). Figure 1c–e show SOT-induced switching measurements of the sample depending on the polarity of $\Delta V_G$. Here, $\Delta V_G$ is the difference in voltages applied to the left ($V_{G,L}$) and right ($V_{G,R}$) gate electrodes with respect to the ground connected to the Pt bottom layer. Without applying a gate voltage, i.e., $\Delta V_G = 0$ ($V_{G,L} = V_{G,R} = 0$ V), the sample shows a typical SOT switching behaviour that occurs only when an in-plane magnetic field ($B_x$) is applied (Fig. 1c);[4,12,13] a positive current favours up-to-down switching under a positive $B_x$. This switching polarity corresponds to a positive spin Hall angle of Pt. Remarkably, when nonzero $\Delta V_G$ is applied, the sample shows a deterministic SOT switching even in the absence of $B_x$ (Fig. 1d, e). Furthermore, the switching polarity is determined by the sign of $\Delta V_G$; a positive current favours up-to-down switching for $\Delta V_G > 0$ ($V_{G,L} = +8$ V, $V_{G,R} = 0$ V), and it is opposite for $\Delta V_G < 0$ ($V_{G,L} = 0$ V, $V_{G,R} = +8$ V). This field-free deterministic SOT switching evidences the $z$-SOT due to $\Delta V_G$ that breaks lateral symmetry. Scanning transmission electron microscopy and electron-energy loss spectroscopy measurements reveal that the $\Delta V_G$ induces the oxygen ion redistribution in oxides depending on its polarity; a larger oxygen ion concentration on the side where a positive bias is applied, compared to the other side (Supplementary Note 1). This might be responsible for the voltage-induced lateral asymmetry. Note that field-free switching is achieved when the $\Delta V_G$ is greater than 8 V, which is a critical $\Delta V_G$ to create a net lateral asymmetry.

We point out that our device has a distinctive advantage that the deterministic SOT switching polarity is electrically controllable. This electrical controllability cannot be achieved with previously reported ones demonstrating $z$-SOT using a hybrid FM/ferroelectric structure[30], a wedged structure[36], a tilted magnetic anisotropy[40,41], a chirally coupled nanomagnets[42], a structural asymmetry[38,43], and a low crystal symmetry material[44,45]. Moreover, this electrical controllability offers programmable logic operations by utilizing the reversible and non-volatile characteristics (Supplementary Note 2). As an example, we demonstrate that XOR and AND logic gate operations are realized in a single device using the gate voltage and input current as two input parameters and the magnetization direction as digital output (See details in Supplementary Note 3). This allows for multi-functional memories or programmable spin logic devices, offering the way for the realization of spin-based logic-in-memory devices.

We next systematically test the $z$-SOT in a Pt/Co/AlO$_x$/TiO$_2$ structure (TiO$_2$ sample) by performing in-plane harmonic measurements[46], in which the 1st and 2nd harmonic Hall resistances ($R_{xy}^{1\omega}, R_{xy}^{2\omega}$) are measured with an a.c. current $I_{ac}$ by rotating the sample (azimuthal angle $\varphi$) under a fixed in-plane magnetic field $\mathbf{B}_{ext}$ (Fig. 2a and Methods). The $R_{xy}^{2\omega}$ is given as

$$R_{xy}^{2\omega}(\varphi) = \left( R_{AHE} \frac{B_{DLT}^y}{B_{eff}} + R_{VT}^{2\omega} \right) \cos\varphi + 2R_{PHE} \frac{B_{FLT}^y + B_{Oe}}{B_{ext}} \left( 2\cos^3\varphi - \cos\varphi \right) \\ - 2R_{PHE} \frac{B_{DLT}^z}{B_{ext}} \cos 2\varphi + R_{AHE} \frac{B_{FLT}^z}{B_{eff}}, \tag{1}$$

where $R_{AHE}$ and $R_{PHE}$ are the anomalous Hall and planar Hall resistances, respectively; $B_{DLT}^y$ ($B_{FLT}^y$) is the damping-like (field-

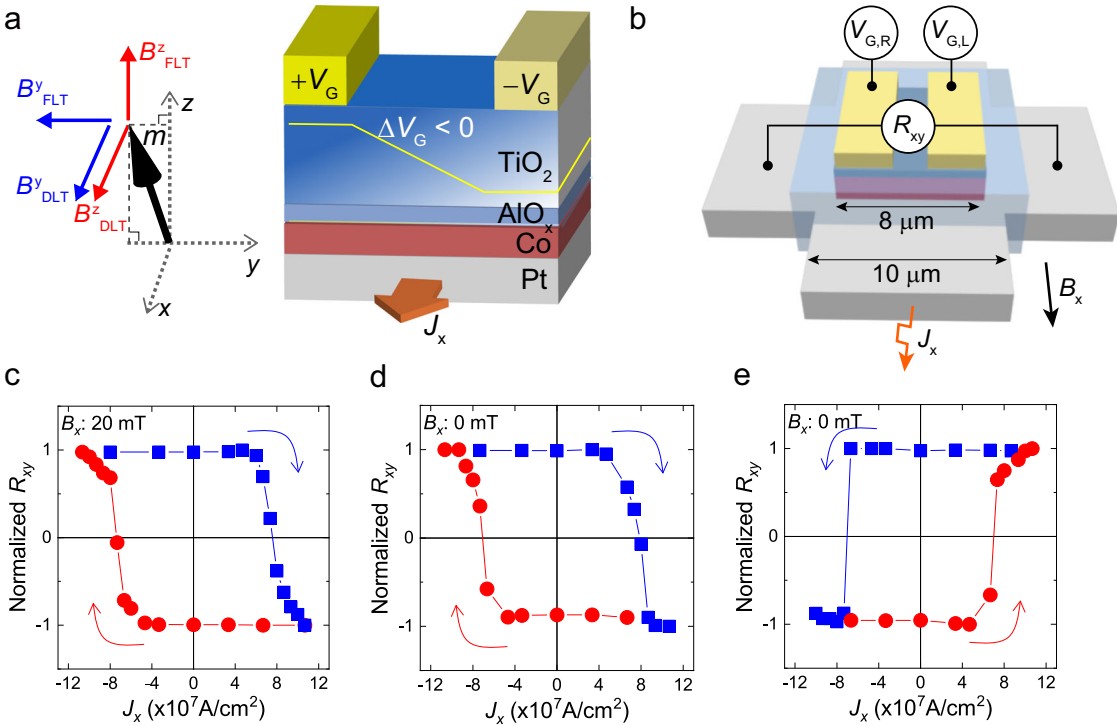

**Fig. 1 Electrical control of field-free SOT switching in Pt/Co/AlO$_x$/TiO2 samples. a** Schematic of the gate voltage-induced lateral symmetry breaking. Gate voltage difference ($\Delta V_G$) induces an electric-field modulation along the $y$-direction, creating additional lateral symmetry-breaking. With a charge current along $x$-direction, this lateral symmetry-breaking generates out-of-plane spin–orbit fields (red arrows); field-like effective field ($B^z_{FLT}$), and damping-like effective field ($B^z_{DLT}$). Those result in additional SOTs ($z$-SOT) in direction of **m** × **z** by $B^z_{FLT}$ and **m** × (**m** × **z**) by $B^z_{DLT}$, where **m** is located in $y$-$z$ plane. The blue arrows indicate the in-plane spin–orbit fields ($B^y_{FLT}$ and $B^y_{DLT}$) induced by symmetry-breaking along the $z$-direction. **b** Schematic illustration of the Hall-bar device with two side gates and the sample structure of Pt/Co/AlO$_x$/TiO$_2$, where the inset shows the optical microscopic image. **c** Current-induced SOT switching for $\Delta V_G = 0$ ($V_{G,L} = V_{G,R} = 0$ V). $B_x = 20$ mT. **d, e** Field-free spin–orbit torque switching for $\Delta V_G > 0$ ($V_{G,L} = +8$ V, $V_{G,R} = 0$ V) (**d**) and the $\Delta V_G < 0$ ($V_{G,L} = 0$ V, $V_{G,R} = +8$ V) (**e**). 8 V corresponds to the electric field of 2.5 MV/cm. Here, the blue (or red) dot arrows indicate from up-to-down (or down-to-up) switching direction.

like) effective field originating from conventional $y$-spin accumulation ($y$-SOT), and $B^z_{DLT}$ ($B^z_{FLT}$) is the damping-like (field-like) effective field additionally generated by $z$-SOT, and $B_{Oe}$ is the Oersted field; $B_{eff}$ is the effective magnetic field, defined as $B_{eff} = B_{ext} + B_{dem} - B_{ani}$, where $B_{dem}$ and $B_{ani}$ are the demagnetization field and the anisotropy field of FM, respectively; $R^{2\omega}_{\nabla T}$ is the anomalous Nernst contribution. Notably, the only $B^z_{DLT}$ has a $\cos 2\varphi$ dependence (Supplementary Note 4), which allows us to unambiguously demonstrate $z$-SOT if generated in the sample. Note that $B^z_{FLT}$ creates angle-independent offset, which is uneasy to identify from harmonic Hall signals.

We examine the sample with four different gate-voltage conditions: ($V_{G,L} = V_{G,R} = +8$ V), ($V_{G,L} = V_{G,R} = -8$ V), ($V_{G,L} = +8$ V and $V_{G,R} = -8$ V), and ($V_{G,L} = -8$ V and $V_{G,R} = +8$ V), which are denoted as $V_G^{(+,+)}$, $V_G^{(-,-)}$, $V_G^{(+,-)}$, and $V_G^{(-,+)}$, respectively. We note that $\Delta V_G = 0$ for both $V_G^{(+,+)}$ and $V_G^{(-,-)}$ while $\Delta V_G > 0$ ($\Delta V_G < 0$) for $V_G^{(+,-)}$ ($V_G^{(-,+)}$). Figure 2b shows the representative measurement data of the $R^{2\omega}_{xy}$ measured at $B_{ext} = 3$ T for those voltage conditions, from which we separate $\cos\varphi$, ($2\cos^3\varphi - \cos\varphi$), and $\cos 2\varphi$ components (Fig. 2c–e). In particular, Fig. 2e shows that the $\cos 2\varphi$ component appears only in the sample under asymmetric gate voltages, i.e., $V_G^{(+,-)}$ and $V_G^{(-,+)}$. We repeat the measurement with a different $B_{ext}$ and plot each component as a function of the $B_{eff}$ (or $B_{ext}$) in Fig. 2f–h. From the slope of the graphs, we extract the asymmetric gate voltage ($= \Delta V_G$) dependence of the $B^y_{DLT}$, $B^y_{FLT}$, and $B^z_{DLT}$ for a current density of $1 \times 10^7$ A/cm$^2$ [Table 1]. Two interesting observations are worth noting; firstly, the $B^y_{DLT}$ and $B^y_{FLT}$ due to

$y$-SOT are enhanced (reduced) when a positive (negative) voltage is applied to both gates: larger $B^y_{DLT}$ and $B^y_{FLT}$ for $V_G^{(+,+)}$ than for $V_G^{(-,-)}$, whereas they are not significantly different for the asymmetric voltage application of $V_G^{(+,-)}$ or $V_G^{(-,+)}$. This result demonstrates that the $y$-SOT can be quantitatively modulated by a symmetric gate voltage [$V_G^{(+,+)}$ and $V_G^{(-,-)}$]. Secondly, and more importantly, a sizable $B^z_{DLT}$ due to $z$-SOT occurs when applying asymmetric voltages ($V_G^{(+,-)}$, $V_G^{(-,+)}$; i.e., nonzero $\Delta V_G$). The magnitude of $B^z_{DLT}$ is about 10% of that of $B^y_{DLT}$. We further confirm the $z$-SOT by performing the hysteresis loop shift experiments of the sample with various $\Delta V_G$'s (Supplementary Note 5), of which results are consistent with those of the SOT switching (Fig. 1) and in-plane harmonic measurements (Fig. 2). We note that the amount of the hysteresis loop shift increases gradually with increasing in-plane current without showing a threshold current that requires to overcome the intrinsic damping[37], indicating the presence of $B^z_{FLT}$, which cannot be clearly identified by the harmonic measurement.

To verify whether the $\Delta V_G$-induced $z$-SOT is general, we investigate another sample of a Pt/Co/AlO$_x$/ZrO$_2$ structure (ZrO$_2$ sample), in which the gate oxide is replaced with ZrO$_2$ while maintaining the same remaining structure. Note that the TiO$_2$ and ZrO$_2$ are widely exploited dielectric materials in resistive memory devices[47] due to their unique material properties; TiO$_2$ has a high oxygen mobility[48] while ZrO$_2$ has a high dielectric constant[49]. Figure 3a–c show SOT switching results of the sample; without gate voltages ($\Delta V_G = 0$), the up-to-down switching occurs for a positive current and a positive $B_x$ (Fig. 3a).

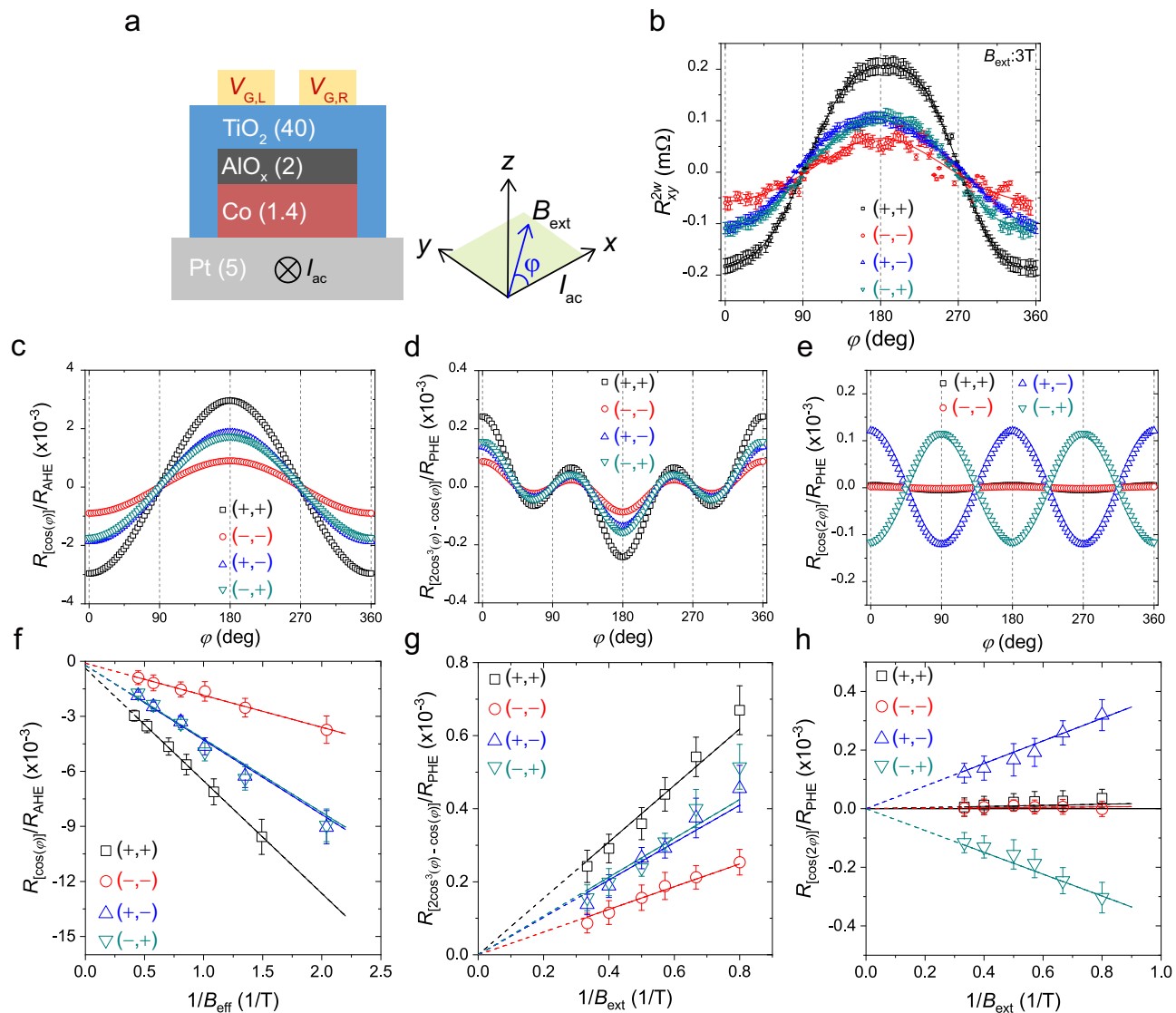

**Fig. 2 Harmonic spin–orbit torque measurements in Pt/Co/AlO$_x$/TiO2 samples. a** Schematic for the measurement configuration. The second harmonic Hall resistances ($R_{xy}^{2\omega}$) for an a.c. current $I_{ac}$ are measured while rotating the sample in the plane (azimuthal angle $\varphi$) under an external field $B_{ext}$. **b** The $R_{xy}^{2\omega}$ versus $\varphi$ curves for the TiO$_2$ samples with four different $V_G$ combinations and $B_{ex} = 3$ T, where the single standard deviation uncertainty of the harmonic Hall voltage measurements is ±0.15 μV, which is included as error bars in the figures. Here, (+,+), (−,−), (+,−), and (−,+) denote ($V_{G,L} = V_{G,R} = +8$ V), ($V_{G,L} = V_{G,R} = −8$ V), ($V_{G,L} = +8$ V and $V_{G,R} = −8$ V), and ($V_{G,L} = −8$ V and $V_{G,R} = +8$ V), respectively. **c–e** The extracted $\varphi$-dependent components of $R_{xy}^{2\omega}$; cos$\varphi$ component (**c**), ($2\cos^3\varphi − \cos\varphi$) component (**d**), and cos$2\varphi$ component (**e**). **f–h** Each $\varphi$-dependent component plotted as a function of $1/B_{eff}$ (or $1/B_{ext}$), where the error bars are due to the uncertainty of the fitting of the $R_{xy}^{2\omega}$ versus $\varphi$ curves to Eq. (1); cos$\varphi$ component versus $1/B_{eff}$ (**f**), ($2\cos^3\varphi − \cos\varphi$) component versus $1/B_{ext}$ (**g**), and cos$2\varphi$ component versus $1/B_{ext}$ (**h**).

| Table 1 Gate voltage dependence of SOT effective fields of the TiO$_2$ sample. | | | | |
|---|---|---|---|---|
| | (+,+) | (−,−) | (+,−) | (−,+) |
| $B_{FLT}^y$ (mT) | 0.77 ± 0.02 | 0.31 ± 0.02 | 0.52 ± 0.02 | 0.52 ± 0.02 |
| $B_{DLT}^y$ (mT) | − 6.13 ± 0.01 | − 1.74 ± 0.01 | − 4.03 ± 0.03 | − 4.03 ± 0.03 |
| $B_{DLT}^z$ (mT) | 0.02 ± 0.002 | 0.01 ± 0.002 | 0.38 ± 0.02 | − 0.38 ± 0.02 |
| Here, the current density is $1 \times 10^7$ A/cm$^2$. | | | | |

The switching polarity is the same with that of the TiO$_2$ sample shown in Fig. 1c. When applying asymmetric electric voltages ($\Delta V_G \neq 0$), the ZrO$_2$ sample also shows deterministic switching similar to the TiO$_2$ sample. It is found that the field-free switching current density ($J_{SW}$) gradually reduces when increasing $\Delta V_G$ (Supplementary Note 6). Furthermore, we examine the

magnetization reversal characteristics during field-free SOT switching using magneto-optical Kerr effect (MOKE) measurement. This indicates that the domain reversal starts at the center of the FM island (or the edge of the gate electrode), where the $\Delta V_G$-induced z-SOT is maximum (Supplementary Note 7). These results unambiguously demonstrate that the generation of z-SOT

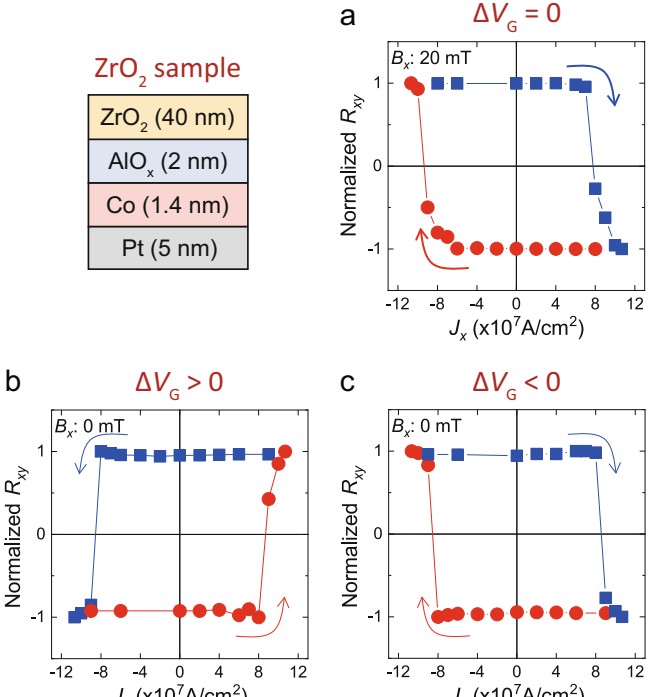

**Fig. 3 Electric-field control of the field-free switching in Pt/Co/AlO$_x$/ ZrO2 samples. a** Current-induced SOT switching for $\Delta V_G = 0$ ($B_x = 20$ mT). **b, c** Field-free SOT switching for $\Delta V_G > 0$ (**b**) and $\Delta V_G < 0$ (**c**). Here, the blue (or red) dot arrows indicate from up-to-down (or down-to-up) switching direction.

is due to $\Delta V_G$. However, unexpectedly, the switching polarity of the ZrO$_2$ sample is opposite from that of the TiO$_2$ sample; a positive (negative) current favours the down-to-up switching for $\Delta V_G > 0$ ($\Delta V_G < 0$). This indicates that the direction of $z$-SOT generated in the ZrO$_2$ sample is opposite to that of the TiO$_2$ sample, which is supported by the additional measurements of $z$-SOT of the ZrO$_2$ samples using in-plane harmonic Hall (Supplementary Note 8) and hysteresis loop shift (Supplementary Note 5) experiments. A possible reason of the opposite polarity depending on the gate oxide will be discussed later.

**Modulation of Rashba effect in ferromagnet/oxide interface.** We now discuss the physical origin of the electric-field-induced $z$-SOT. The first possible cause is the lateral modulation of PMA through voltage-controlled magnetic anisotropy (VCMA) effect[36,40,41]. The asymmetric gate voltages give rise to a gradient of PMA along the $y$-direction of which sign depends on the sign of $\Delta V_G$, resulting in a field-free switching. To test this possibility, we measure the VCMA effect of the TiO$_2$ and ZrO$_2$ samples with four different gate-voltage conditions of $V_G^{(+,+)}$, $V_G^{(-,-)}$, $V_G^{(+,-)}$, and $V_G^{(-,+)}$ (Supplementary Note 9). We find an enhancement (reduction) in PMA for both samples with a gate voltage of $V_G^{(-,-)}$ ($V_G^{(+,+)}$) and no significant variation in PMA for the $V_G^{(+,-)}$ and $V_G^{(-,+)}$ conditions. Notably, the polarity of the VCMA effect is the same for both the TiO$_2$ and ZrO$_2$ samples and thus $\Delta V_G$–induced PMA gradient along the $y$-direction cannot explain the opposite signs of the $z$-SOT. As a result, the VCMA effect is excluded as a cause of $z$-SOT. We also perform COMSOL simulations to check the gate voltage dependence of the current distribution (Supplementary Note 10), which allows us to rule out the possible contributions of the lateral current distribution.

Another possibility is the lateral variation of the Rashba effect at the Co/oxide interface, which can also be induced by asymmetric gate voltages. The magnitude of Rashba effect is proportional to the built-in electric-field originating from the band structure at the Co/oxide interface, which we estimate by measuring the potential barrier height[50] in Co (10 nm)/AlO$_x$ (2 nm)/gate oxide (5 nm)/Ru (20 nm) tunnel junctions depending on gate voltage. Note that we use a 5 nm gate oxide that is thin enough for electrons to tunnel through. Figure 4a, b show the $I$–$V$ characteristics of the tunnel junctions, demonstrating that the change in the potential barrier by a gate voltage ($V_G$) depends on the gate oxide. For the junction with a TiO$_2$, after applying a positive (negative) $V_G$, the tunnelling current increases (decreases) (Fig. 4a), indicating that the potential barrier diminishes (increases). In contrast, the junction with a ZrO$_2$ shows the reverse electric-field effect; a positive $V_G$ decreases the tunnel current and increases the potential barrier (Fig. 4b). The inset of Fig. 4a, b schematically illustrates the lateral variation of the barrier height ($\phi$) for the gate oxide. The opposite electric-field effect may be due to the different transport mechanisms of the oxides; oxygen ion migration (charge trap) is the dominant mechanism in TiO$_2$ (ZrO$_2$)[47–49]. However, further investigation is required to clarify the gate oxide dependence of the electric-field effect. Nonetheless, the above results support the hypothesis that the $z$-SOT results from the lateral modulation of the built-in electric-field and associated Rashba effect at the Co/oxide interface. The asymmetric gate voltage ($\Delta V_G$) induces the variation of the potential barrier along the $y$-direction, leading to the generation of $z$-SOT. This result explains the different polarity of $z$-SOT between the TiO$_2$ and ZrO$_2$ samples.

This scenario is further supported by in-plane harmonic Hall measurement of a Pt (0.5 nm)/Co (2 nm)/AlO$_x$ (2 nm)/gate oxide (40 nm) samples, in which FM Co is fully covered by a single gate. Figure 4c, d show the $(2\cos^3 \varphi - \cos\varphi)$ component of $R_{xy}^{2\omega}$ as a function $1/B_{ext}$ depending on $V_G$, which demonstrates that the change in $B_{FLT}^y$ with $V_G$ depends on gate oxide; for the sample with TiO$_2$ (ZrO$_2$) gate oxide, the $B_{FLT}^y$ increases (decreases) by a positive $V_G$. Since the SHE is negligible in a 0.5 nm Pt (Supplementary Note 11), the $B_{FLT}^y$ primarily originates from the Pt/Co and/or Co/AlO$_x$ interfaces. Furthermore, the opposite electric-field effect of $B_{FLT}^y$ between the TiO$_2$ and ZrO$_2$ samples suggests that the variation of the $B_{FLT}^y$ can be attributed to the electric-field-controlled Rashba effect of the Co/AlO$_x$ interface. This is consistent with the trend of the electric-field-controlled potential barrier height, supporting that the lateral variation of the Rashba effect is the key element for the generation of the electric-field-induced $z$-SOT. It is reported that spin scattering at the FM/oxide interface can contribute to field-like torque[51]. This interfacial spin scattering is consistent with the Rashba effect at the FM/oxide interface, which causes the interfacial spin–orbit precession[37,52] and associated field-like torques[53].

**Theoretical and numerical demonstration of electric-field-induced $z$-SOT.** We now present that the lateral modulation of Rashba effect, which is induced by the lateral oxygen gradient, is the key element for the $z$-SOT by means of a symmetry argument and first-principles calculations. First, we present a general symmetry argument. The $z$-SOT $\left.\frac{d\mathbf{m}}{dt}\right|_\perp$ can be described as

$$\left.\frac{d\mathbf{m}}{dt}\right|_\perp = aJ\mathbf{m} \times \mathbf{z} + bJ\mathbf{m} \times (\mathbf{m} \times \mathbf{z}), \qquad (2)$$

where $J$ is the current density applied along the $x$-direction, $\mathbf{m}[= (m_x, m_y, m_z)]$ is the direction of magnetization, $\mathbf{z}$ is the direction normal to the film plane, and $a$ ($b$) represents the

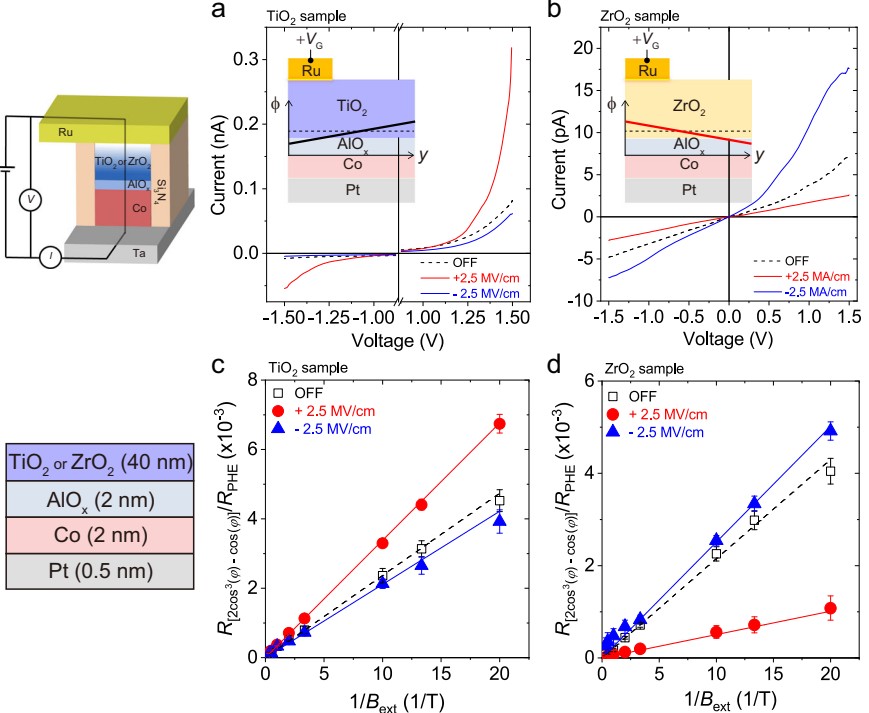

**Fig. 4 Voltage-induced variation of the potential barrier and field-like torque. a, b** $I$–$V$ characteristics in the Ta (10 nm)/Co (10 nm)/AlO$_x$ (2 nm)/TiO$_2$ (5 nm)/Ru (20 nm) (**a**) and Ta (10 nm)/Co (10 nm)/AlO$_x$ (2 nm)/ZrO$_2$ (5 nm)/Ru (20 nm) tunnel junctions (**b**) depending on pre-biased gate voltages $V_G$ corresponding to ±2.5 MV/cm. The inset corresponds to schematic drawing of the lateral variation of barrier height ($\phi$) of the gate oxide, where the black and red line indicate the potential barrier height of oxide layer. **c, d** Field-like SOT component of depending on pre-biased gate voltage $V_G$ corresponding to ±2.5 MV/cm for Pt (0.5 nm)/Co (2 nm)/AlO$_x$ (2 nm)/TiO$_2$ (40 nm) (**c**) and Pt (0.5 nm)/Co (2 nm)/AlO$_x$ (2 nm)/ZrO$_2$ (40 nm) (**d**) samples. The error bars are due to the uncertainty of the fitting of the $R_{xy}^{2\omega}$ versus $\varphi$ curves to Eq. (1).

magnitude of field-like (damping-like) component of the $z$-SOT. The mirror reflection $\mathcal{M}_y$ of Eq. (2) with respect to the $xz$ plane corresponds to the following operations: $m_x \rightarrow -m_x$, $m_y \rightarrow m_y$, $m_z \rightarrow -m_z$, $\frac{dm_x}{dt} \rightarrow -\frac{dm_x}{dt}$, $\frac{dm_y}{dt} \rightarrow \frac{dm_y}{dt}$, $\frac{dm_z}{dt} \rightarrow -\frac{dm_z}{dt}$, and $J \rightarrow J$. This mirror reflection results in $\frac{d\mathbf{m}}{dt}\big|_\perp = -\frac{d\mathbf{m}}{dt}\big|_\perp$, meaning that the $z$-SOT is absent when the $\mathcal{M}_y$ symmetry is preserved. However, the $z$-SOT is allowed when the $\mathcal{M}_y$ symmetry is broken[34]. In our experiment, the $\mathcal{M}_y$ symmetry is broken by the lateral gradient of Rashba effect.

We next present our first-principles results. We numerically demonstrate that the lateral oxygen gradient ($\nabla V_o$) results in the $z$-component non-equilibrium spin density, corresponding to the $z$-SOT. To this end, we compute a non-equilibrium spin density in Pt/Co/O structures with and without $\nabla V_o$, based on the linear response theory within density functional theory (Supplementary Note 12). Calculated non-equilibrium spin density $\delta\mathbf{s}[=(\delta s_x, \delta s_y, \delta s_z)]$ for Pt/Co/O structures are shown in Fig. 5. In our calculation, the electric field is applied along the $x$-axis and the magnetization $\mathbf{m}$ of Co is aligned in the $y$-axis. The results show that the magnitudes of $\delta s_y$ are similar for both structures with and without $\nabla V_0$ (Fig. 5d). In contrast, the magnitudes of $\delta s_x$ and $\delta s_z$ are nonzero only in the structure with $\nabla V_0$ (Fig. 5c, e). Since the SOT is proportional to $\mathbf{m}\delta\mathbf{s}$ and $\mathbf{m}$ is aligned in the $y$-direction, $\delta s_x$ corresponds to the (**mz**)-component of $\delta\mathbf{s}$ whereas $\delta s_z$ corresponds to the **z**-component of $\delta\mathbf{s}$. Comparing to Eq. (2), one finds that $\delta s_x$ and $\delta s_z$ correspond to $b$ and $a$, respectively. These results confirm that the lateral oxygen gradient, which induces the lateral modulation of Rashba effect (Supplementary Note 12), makes both damping-like and field-like $z$-SOT. Furthermore, it is also found that $\delta s_z$ is linearly

proportional to the $\nabla V_o$, indicating $\delta s_z$ is always nonzero once (even small) oxygen gradient is present (Supplementary Note 13), and that $\delta s_z$ is the same for the same oxygen gradient regardless of the system size, demonstrating that devices properties would be maintained even in the nanometer scale devices (Supplementary Note 14).

## Discussion

We demonstrate the generation of $z$-SOT in Pt/Co/AlO$_x$ structures by inducing a lateral voltage gradient using two side gates, which enables field-free switching of perpendicular magnetization and electrical control of the switching polarity. We also show that the magnitude and direction of SOT is electrically controllable in such structures. Furthermore, we find that the $z$-SOT depends on gate oxides that exhibit an opposite electric-field effect on the potential barrier height while maintaining the same voltage-controlled magnetic anisotropy effect, indicating that the lateral modulation of Rashba effect at the Co/oxide interface by an electric voltage is a key ingredient of the generation of $z$-SOT.

We finally discuss the feasibility of our lateral gate SOT device. Our device performances such as speed and endurance depend on voltage-induced oxygen migration. In this study, we used a 40 nm thick gate oxide for which a gate voltage ($V_G$) needs to be applied for 5 min at 100 °C to observe a noticeable electric-field effect. This is not a fundamental limitation of the operation speed of our device, but it can be improved by materials engineering. By introducing a double gate oxide of TiO$_2$(2 nm)/ZrO$_2$(5 nm) structure, we show that the gate operation works at room temperature with a short voltage pulse of 20 µs (Supplementary Note 15) and high endurance (Supplementary Note 16). Furthermore, we perform SPICE circuit simulations under the

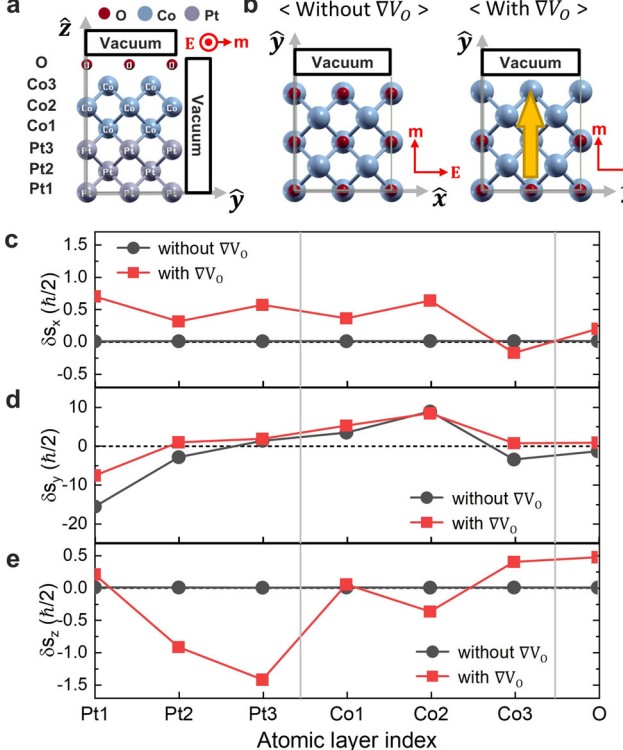

**Fig. 5 Non-equilibrium spin density for the structures with and without the lateral oxygen gradient. a, b** Schematics of the two different Pt/Co/O structures in $yz$-plane (side view) (**a**) and $xy$-plane (top view) (**b**). In **b**, the left panel shows the structure without the lateral oxygen gradient ($\nabla V_o$), and the right panel shows the structure with the lateral oxygen gradient. The direction of lateral symmetry breaking is indicated by yellow arrow. In both **a** and **b**, the directions of magnetization (**m**) and electric field (**E**) are denoted by red arrows. **c–e** Atomic layer resolved non-equilibrium spin densities $\delta s_x$ (**c**), $\delta s_y$ (**d**), and $\delta s_z$ (**e**) for the Pt/Co/O structure with and without lateral oxygen gradient.

assumption that our technique, a field-free SOT switching device with a side gate, can be applicable to nanosized magnetic tunnel junctions (Supplementary Note 17), which demonstrate the potential of implementing an XOR gate based on our SOT device with reduced device area and comparable energy consumption compared to CMOS-based XOR gates.

## Methods

**Sample preparation**. The samples of Ta (2 nm)/Pt (5 nm)/Co (1.4 nm)/AlO$_x$ (2 nm) structures were deposited on Si/Si$_3$N$_4$ wafer using d.c. magnetron sputtering with a base pressure of less than $4.0 \times 10^{-6}$ Pa at room temperature. All metallic layers were grown with a working pressure of 0.4 Pa and a power of 30 W, while the AlO$_x$ layer was fabricated by deposition of an Al layer and subsequent plasma oxidation with an O$_2$ pressure of 4.0 Pa and a power of 30 W for 75 s. The samples were then covered by a gate oxide of TiO$_2$ or ZrO$_2$ (40 nm) that is grown at 125 °C by plasma enhanced atomic layer deposition (PE-ALD) using TDMAT [Tetrakis (dimethylamido) titanium] or TEMAZ [Tetrakis(ethylmethylamido)zirconium] and O$_2$ precursors. The oxygen plasma was formed with a rf power of 60 W and a flow of 500 sccm O$_2$. The Hall-bar-structure devices with a $10 \times 10$ µm cross that includes a square-shaped ferromagnetic island with $8 \times 8$ µm were fabricated using photolithography and Ar ion-milling. Two gate electrodes on both sides of the ferromagnetic island with a spacing of 2 µm were fabricated by deposition of Ru (50 nm) and subsequent lift-off technique.

**Gate voltage application**. Prior to the electrical measurements, a gate voltage equivalent to ±2.5 MV/cm was applied to the Ru gate electrode for 5 min at 100 °C with respect to the ground connected to the bottom Pt layer unless otherwise specified. Thereafter, the spin–orbit torque measurements were conducted at room temperature with the gate electrode floating.

**Spin-orbit torque measurements**. Spin-orbit torque was characterized using an in-plane harmonic lock-in technique. The $R_{xy}^{1\omega}$ and $R_{xy}^{2\omega}$ for an a.c. current $I_{ac}$ of 11 Hz were simultaneously measured while rotating the sample in the plane (azimuthal angle $\varphi$) under an external field $B_{ext}$. The magnitude of $B_{ext}$ is larger than the anisotropy field $B_{ani}$ of the sample, so the magnetization is aligned parallel to the direction of $B_{ext}$. The single standard deviation uncertainty of the harmonic Hall voltage measurements is ±0.15 µV, which is included as error bars in the figures. The SOT-induced switching experiments were performed by measuring the $R_{AHE}$ using a d.c. current of 100 µA after applying a current pulse of 70 µs with (or without) an in-plane magnetic field $B_x$. The single standard uncertainty of the Hall resistance measurements is ±1 mΩ. Corresponding error bars are included in the figures. In most case, they are much smaller than the size of symbols.

All measurements were conducted at room temperature. More than three samples are measured for each type of sample; data are qualitatively reproducible.

**I–V curve measurements**. The $I–V$ characteristics of the sample were measured while sweeping a dc voltage between ±1.5 V at room temperature using a Ta (10 nm)/Co (10 nm)/AlO$_x$ (2 nm)/gate oxide (5 nm)/Ru (20 nm) tunnel junction with a junction area of $8 \times 8$ µm. The top Ru electrode was biased and the bottom Ta was grounded, so that a positive (negative) voltage corresponds to an electric-field pointing down (up). Prior to the $I–V$ measurement, a gate voltage equivalent to 2.5 MV/cm was applied to the top Ru gate electrode for 5 min at 100 °C.

## Data availability

The data that support the findings of this study are available from the corresponding author upon reasonable request.

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

## Acknowledgements

We acknowledge discussion with Hyun-Woo Lee. B.-G.P., K.-J.L., & J.P. acknowledge support from Samsung Research Funding Center of Samsung Electronics under Project Number SRFC-MA1802-01. K.-W.K. acknowledges support from the KIST Institutional program.

## Author contributions

The study was performed under the supervision of B.-G.P. J.-G.C. designed the experiment. M.-G.K. and J.-G.C. fabricated devices and performed the spin–orbit torque switching and in-plane harmonic measurements and M.-G.K. tested the voltage-controlled potential barrier via *I–V* measurement. M.-G.K., J.-G.C., and B.-G.P. performed data analysis with the help of H.-J.P. and K.-J.L. J.J. helps to fabricate the gate oxide using ALD. J.Y.P. and J.M.Y. analyzed the TEM and EELS measurements. H.-J.P., K.-W.K., J.H.O., and K.-J.L. performed theoretical study. T.L. and K.-J.K. help to measure MOKE imaging to examine domain reversal. T.K and J.P. performed SPICE circuit simulation. D.D.V. and J.-R.J. performed COMSOL simulation. M.-G.K., J.-G.C., K.-J.L., and B.-G.P. wrote the manuscript.

## Competing interests

The authors declare no competing interests.
