## [Peer Review File · Nature Communications]

Reviewers' Comments:

Reviewer #1:

Remarks to the Author:

The authors have addressed most of the concerns raised by the referees. Although I still have my doubts regarding the possibility of scaling this device architecture down to nano size, the authors have shown its potential. I believe that the present manuscript will be a good fit for *Nature Communications*.

Reviewer #2:

Remarks to the Author:

I think this work has several important findings attracting the attentions of specialists in this field; for example (i) ferromagnet/oxide interface has sizable contribution to the magnetization dynamics induced by SOT and (ii) lateral modulation of the gate state generates SOT along the film normal direction (z-SOT), resulting in field-free magnetization switching. Therefore, I think this manuscript in principle warrant publication in *Nature Communications*. However, before the acceptance, the following three issues should be addressed.

(1) Contribution of bulk Pt, Pt/Co interface, and Co/AlO_x interface for SOT should be clarified. It is likely from the presented results that Co/AlO_x plays a crucial role for z-SOT; however, it is not clear how the bulk Pt, Pt/Co interface, and Co/AlO_x interface contribute to the conventional in-plane SOT. In addition, I think the authors should consider the effect reported in PRB 94, 140414(R) (2016), in which fieldlike torque generated through a spin scattering at FM/oxide interface is reported. I am wondering if this kind of effect can also explain the present results even without considering the Rashba effect.

(2) While the authors state that the z-SOT and associated switching polarity are controllable by gate voltage in a reversible and "non-volatile" manner, but the "non-volatile" control is not fully supported in the present manuscript. How long does the state of oxide (TiO₂ or ZrO₂) last? The authors should either show evidence of "non-volatility" or tone down this claim.

(3) The results shown in Fig. 4 clearly indicates that TiO₂ and ZrO₂ have different properties with each other, resulting in an opposite switching polarity shown in Fig. 1d,e and Fig. 3b,c, and the authors attribute this result to different transport mechanism of the two oxides (ion migration/charge trap) in lines 184-186. However, I think the actual picture happening in the oxide in their side-gate device is not clear to the readers and should be schematically shown in Fig. 4 or SI. Do the authors speculate that the electric charges with lateral gradient remain after turning off the side-gate voltage, and the direction of gradient differs between TiO₂ and ZrO₂? Such illustration should aid in the understanding of potential readers.

Reviewer #3:

Remarks to the Author:

The authors have given further quantitative estimates on the oxygen gradients which was one of my concerns, albeit still far from what is observed in the measured devices. They however argue that any gradient, how small what so-ever, will lead to field free switching. Which in essence for the physics symmetry breaking might be true. However, in real devices and in the atomic world where gradients will be discrete this is not true. Hence, as the authors claim real-life applications should give a lower bound on dependence of the 'probability' of field-free switching when a small gradient is present. In a sense saying that it will always work, is weakening the claim of the paper, as the

observed result just as well be explained due to a minor tilt of the sample during e.g. sputtering or processing or wherever.

The scaling claim in SPICE is still not sufficiently accompanied by a warning. Give a clear limit at which the effect here has been shown, and indicate that if, and only if, the scaling holds SPICE can be used accordingly, preferably even add a lower limit with an argument. Again, too many scaling problems are created by uninformed engineers scaling down such effects to $10 \times 10 \text{ nm}^2$ and claiming it will work as a charm.

Overall I am still in favor of publishing this work. The limits are clear, time effects are beyond the scope but addressed.

Reviewer #1 (Remarks to the Author)

The authors have addressed most of the concerns raised by the referees. Although I still have my doubts regarding the possibility of scaling this device architecture down to nano size, the authors have shown its potential. I believe that the present manuscript will be a good fit for Nature Communications.

Response) We appreciate the reviewer's recommendation for publication of our manuscript in *Nature Communications*.

Reviewer #2 (Remarks to the Author)

I think this work has several important findings attracting the attentions of specialists in this field; for example (i) ferromagnet/oxide interface has sizable contribution to the magnetization dynamics induced by SOT and (ii) lateral modulation of the gate state generates SOT along the film normal direction (z-SOT), resulting in field-free magnetization switching. Therefore, I think this manuscript in principle warrant publication in Nature Communications. However, before the acceptance, the following three issues should be addressed.

Response) We appreciate the reviewer's comment that “*this work has several important findings attracting the attentions of specialists in this field; I think this manuscript in principle warrant publication in Nature Communications.*” We respond to the reviewer’s additional comments below, which hopefully alleviates the reviewer’s concerns and the revised manuscript is now acceptable for publication.

(1) Contribution of bulk Pt, Pt/Co interface, and Co/AlO_x interface for SOT should be clarified. It is likely from the presented results that Co/AlO_x plays a crucial role for z-SOT; however, it is not clear how the bulk Pt, Pt/Co interface, and Co/AlO_x interface contribute to the conventional in-plane SOT. In addition, I think the authors should consider the effect reported in PRB 94, 140414(R) (2016), in which field-like torque generated through a spin scattering at FM/oxide interface is reported. I am wondering if this kind of effect can also explain the present results even without considering the Rashba effect.

Response) We appreciate the reviewer’s comment that the contributions of bulk Pt, Pt/Co interface, and Co/AlO_x interface to in-plane spin-orbit torques (SOTs) should be examined. In this study, we performed in-plane SOT measurements of two samples with different Pt thicknesses: Pt (5 nm)/Co (1.4 nm)/AlO_x (2 nm) [Fig. 2 of the main text] and Pt (0.5 nm)/Co (2 nm)/AlO_x (2 nm) structures [Fig. 4 of the main text and Supplementary Note 10]. These measurements show that the spin Hall effect is negligible in the sample with a thin Pt (0.5 nm). Two points are worth noting; first, the sample with a thick Pt (5 nm) shows a sizable damping-like SOT effective field (B_{DLT}^y) and electric controllability (Fig. 2f in the original main text and Fig. S7a in the Supplementary Note 7). In contrast, the samples with a thin Pt (0.5 nm) exhibits negligible B_{DLT}^y (Supplementary Note 10 in the original manuscript). This result demonstrates that the B_{DLT}^y dominantly arises from the bulk Pt or spin Hall effect (SHE). Second, we observed non-trivial field-like SOT effective field (B_{FLT}^y) in the sample with a thin Pt (0.5 nm) [Figs. 4c,d in the original main text]. This indicates that the B_{FLT}^y originates from the Pt/Co and/or Co/AlO_x interfaces. At this stage, we cannot quantitatively separate the contribution of the Pt/Co and Co/AlO_x interfaces to B_{FLT}^y ; however, the opposite sign of the voltage-induced modulation of B_{FLT}^y in the samples with different gate oxides [Figs. 4c,d of the original main text] indicates that

the Rashba effect of the Co/AlO_x interface plays an important role in determining B_{FLT}^y .

Next, we discuss whether our results can be explained by the field-like torque (FLT) mechanism reported in *PRB 94, 140414 (R) (2016)*, where FLT is generated by spin scattering at the FM/oxide interface. We agree with the reviewer that the spin scattering at the FM/oxide interface has an important role in FLT. To our understanding, however, this spin scattering mechanism is closely related to the Rashba effect at the same FM/oxide interface. A theory paper [PRB 103, 134405 (2021)] investigated the role of the FM/oxide interface in spin-orbit torques and found that the Rashba effect at this interface gives rise mainly to FLT. In this theory, the spin scattering at the FM/oxide interface is the spin-orbit precession scattering at the interface, where the spins injected to the FM/oxide interface undergoes the precession around the effective interfacial Rashba spin-orbit field [Nat. Mater. 17, 509 (2018); PRL 121, 136805 (2018)]. Therefore, the interfacial spin scattering quoted by the reviewer and the Rashba effect in our manuscript are not mutually exclusive but rather closely related each other.

In the revised manuscript on page 10, we added the following sentences regarding the above discussion, “*the B_{FLT}^y primarily originates from the Pt/Co and/or Co/AlO_x interfaces. Furthermore, the opposite electric field effect of B_{FLT}^y between the TiO₂ and ZrO₂ samples suggests that the variation of the B_{FLT}^y can be attributed to the electric field-controlled Rashba effect of the Co/AlO_x interface*” and “*It is reported that spin scattering at the FM/oxide interface can contribute to field-like torque [PRB 94, 140414 (R) (2016)]. This interfacial spin scattering is consistent with the Rashba effect at the FM/oxide interface, which causes the interfacial spin-orbit precession [Nat. Mater. 17, 509 (2018); PRL 121, 136805 (2018)] and associated field-like torques [PRB 103, 134405 (2021)].*”

(2) While the authors state that the z-SOT and associated switching polarity are controllable by gate voltage in a reversible and “nonvolatile” manner, but the “nonvolatile” control is not fully supported in the present manuscript. How long does the state of oxide (TiO₂ or ZrO₂) last? The authors should either show evidence of “non-volatility” or tone down this claim.

Response) We appreciate the reviewer’s comments on the non-volatility of the electric-field effect. To demonstrate the nonvolatile behavior of our device, we examined how long the voltage-controlled magnetic anisotropy (VCMA) effect persists in a Pt (5 nm)/Co (1.4 nm)/AlO_x (2 nm)/TiO₂ (40 nm) structure, which is the same sample used in Fig. 1 of the main text. **Figure R1a** shows the VCMA effect of the sample; coercivity (B_c) is reduced when a V_G of +8V (equivalent to 2.5 MV/cm) is applied. To demonstrate the retention of the VCMA effect, we repeatedly measured the B_c over a week, where the measurement was performed every hour on the first day and once a day thereafter. **Figure R1b** shows that the reduced B_c remains almost the same over the measurement time, demonstrating the VCMA effect remains nearly the same for more than 6×10^5 seconds, corresponding to seven days.

We included the results of the nonvolatile VCMA effect in Supplementary Note 2.

Figure R1. Non-volatility of voltage-controlled magnetic anisotropy in the Pt/Co/AlO_x/TiO₂ sample. (a) Normalized anomalous Hall resistance (R_{xy}) versus out-of-plane magnetic field (B_z) before and after applying $V_G = +8$ V. (b) Coercivity (B_c) versus measurement time.

(3) *The results shown in Fig. 4 clearly indicates that TiO₂ and ZrO₂ have different properties with each other, resulting in an opposite switching polarity shown in Fig. 1d,e and Fig. 3b,c, and the authors attribute this result to different transport mechanism of the two oxides (ion migration/charge trap) in lines 184-186. However, I think the actual picture happening in the oxide in their side-gate device is not clear to the readers and should be schematically shown in Fig. 4 or SI. Do the authors speculate that the electric charges with lateral gradient remain after turning off the side-gate voltage, and the direction of gradient differs between TiO₂ and ZrO₂? Such illustration should aid in the understanding of potential readers.*

Response) Thank you for the comment. As the reviewer commented, the I - V characteristics shown in Figs. 4a,b demonstrate that the change in the potential barrier by a gate voltage (V_G) depends on the gate oxide. The potential barrier height decreases (increases) when applying a positive V_G for the sample with a TiO₂ (ZrO₂). These results suggest that the asymmetric gate voltage (ΔV_G) induces the lateral variation of the potential barrier along the y -direction, leading to the generation of z -SOT. The opposite direction of the potential barrier gradient accounts for the different polarities of z -SOT between the TiO₂ and ZrO₂ samples. We include a schematic of the lateral variation in barrier height (ϕ) as the inset of Fig. 4 in the revised manuscript, which is shown in **Fig. R2**, where the black or red line indicates the potential barrier height of the oxide layer. We believe that the opposite electric field effect may be due to the different transport mechanisms of the oxides; oxygen ion migration (charge trap) is the dominant mechanism in TiO₂ (ZrO₂). However, further investigation is required to clarify the gate oxide dependence of the electric field effect.

Figure R2. Modified Figure 4 of the revised manuscript, where the inset shows a schematic drawing of the lateral variation in barrier height (ϕ) of the gate oxide. The black or red line indicates the potential barrier height of the oxide layer.

Reviewer #3 (Remarks to the Author)

The authors have given further quantitative estimates on the oxygen gradients which was one of my concerns, albeit still far from what is observed in the measured devices. They however argue that any gradient, how small what so-ever, will lead to field free switching. Which in essence for the physics symmetry breaking might be true. However, in real devices and in the atomic world where gradients will be discrete this is not true. Hence, as the authors claim real-life applications should give a lower bound on dependence of the 'probability' of field-free switching when a small gradient is present. In a sense saying that it will always work, is weakening the claim of the paper, as the observed result just as well be explained due to a minor tilt of the sample during e.g. sputtering or processing or wherever.

Response) We appreciate the reviewer's comment. As the reviewer pointed out, real samples may have a gradient that arises during fabrication. However, this gradient is randomly distributed so that it cannot cause a well-defined net gradient along the y-direction transverse to the current direction, which is required for the generation of z-SOT and associated field-free switching. In our experiment, field-free switching is realized when an asymmetric gate voltage (ΔV_G) is larger than 8 V, which is a critical ΔV_G that overcomes unintended non-uniformities to create a net gradient.

In the revised manuscript on page 5, we added the following sentence describing a critical asymmetric gate voltage for field-free switching, "*Note that field-free switching is achieved when the ΔV_G is greater than 8 V, which is a critical ΔV_G to create a net lateral asymmetry.*"

The scaling claim in SPICE is still not sufficiently accompanied by a warning. Give a clear limit at which the effect here has been shown, and indicate that if, and only if, the scaling holds SPICE can be used accordingly, preferably even add a lower limit with an argument. Again, too many scaling problems are created by uninformed engineers scaling down such effects to $10 \times 10 \text{ nm}^2$ and claiming it will work as a charm.

Response) We feel sorry for the insufficient description of the SPICE simulation, and we agree with the reviewer that the simulation results are meaningful only if the scaling holds down to nano-devices. Therefore, we modified the following sentence by including the assumption of the SPICE simulation on page 12 of the revised manuscript; "*We perform SPICE circuit simulations under the assumption that our technique, a field-free SOT switching device with a side gate, can be applicable to nanosized magnetic tunnel junctions (Supplementary Note 17), which demonstrate the potential of implementing an XOR gate based on our SOT device with reduced device area and comparable energy consumption compared to CMOS-based XOR gates*".

Overall I am still in favor of publishing this work. The limits are clear, time effects are beyond the scope but addressed.

Response) We appreciate the reviewer's acknowledgment, "*Overall, I am still in favor of publishing*

this work.” We have responded to the reviewer’s additional comments above, which hopefully convinces her/him to support the publication of our manuscript.

Reviewers' Comments:

Reviewer #2:

Remarks to the Author:

In the previous round, I made three comments, to which the authors have addressed in this revision. I think the first comment on the origin of spin-orbit torque and the second comment on the nonvolatility have been satisfactorily addressed. On the other hand, the authors' action to my third comment is different from what I intended. While the authors added a lateral variation in potential barrier in Fig. 4, I thought that what is actually happening inside the gate oxide should be illustrated, for example, by drawing a possible spatial variation of oxygen owing to the difference in the transport mechanism, because this is the most important ingredient achieving the field-free spin-orbit torque switching in this work. However, if the authors want to keep it vague, I do understand. Apart from that, I found one typo in line 279; a square after "125" should read degree Celsius. Once the above issues are addressed, I would be happy to recommend the acceptance of this manuscript.

Reviewer #2 (Remarks to the Author)

In the previous round, I made three comments, to which the authors have addressed in this revision. I think the first comment on the origin of spin-orbit torque and the second comment on the nonvolatility have been satisfactorily addressed. On the other hand, the authors' action to my third comment is different from what I intended. While the authors added a lateral variation in potential barrier in Fig. 4, I thought that what is actually happening inside the gate oxide should be illustrated, for example, by drawing a possible spatial variation of oxygen owing to the difference in the transport mechanism, because this is the most important ingredient achieving the field-free spin-orbit torque switching in this work. However, if the authors want to keep it vague, I do understand. Apart from that, I found one typo in line 279; a square after "125" should read degree Celsius. Once the above issues are addressed, I would be happy to recommend the acceptance of this manuscript.

Response) We thank the reviewer for carefully reading our responses. We are happy to hear that “*Once the above issues are addressed, I would be happy to recommend the acceptance of this manuscript.*” Following the reviewer’s suggestions, we revised the manuscript as described below.

First of all, we apologize for misunderstanding the reviewer’s comment about the illustration of actual happening inside the gate oxide. We note that the microscopic origin of field-free switching is the lateral variation of the potential barrier and associated Rashba effect at the Co/oxide interface. We, therefore, added corresponding schematics in Figure 4 of the revised manuscript. The reviewer wanted us to explain the different field-free switching polarities between the samples with TiO₂ and ZrO₂ gate oxides based on the transport mechanism of the oxide. In the original manuscript, we speculated that the opposite electric field effect might be due to the different transport mechanisms of the oxides, but this has not been confirmed by experimental or theoretical studies. Further investigation of the oxygen distribution along the thickness direction is required to address this conclusively. Therefore, we think it would be better not to include schematic drawings in the current manuscript, which may mislead the readers.

Furthermore, we revised the typo in line 279 as “125 °C”.